# The Treatment of Medication-Related Osteonecrosis of the Jaw (MRONJ): A Systematic Review with a Pooled Analysis of Only Surgery versus Combined Protocols

**DOI:** 10.3390/ijerph18168432

**Published:** 2021-08-10

**Authors:** Olga Di Fede, Federica Canepa, Vera Panzarella, Rodolfo Mauceri, Carmine Del Gaizo, Alberto Bedogni, Vittorio Fusco, Pietro Tozzo, Giuseppe Pizzo, Giuseppina Campisi, Antonio Galvano

**Affiliations:** 1Department of Surgical, Oncological and Oral Sciences (Di.Chir.On.S.), University of Palermo, 90127 Palermo, Italy; odifede@odonto.unipa.it (O.D.F.); rodolfo.mauceri@unipa.it (R.M.); delgaizocarmine@gmail.com (C.D.G.); giuseppe.pizzo@unipa.it (G.P.); campisi@odonto.unipa.it (G.C.); antonio.galvano@unipa.it (A.G.); 2Operative Unit of Dentistry and Stomatology, Azienda Ospedaliera Ospedali Riuniti Villa Sofia Cervello, 90146 Palermo, Italy; canepafederica86@gmail.com (F.C.); tozzo.pietro@libero.it (P.T.); 3Department of Biomedical and Dental Sciences and Morphofunctional Imaging, University of Messina, 98124 Messina, Italy; 4Department of Dental Surgery, Faculty of Dental Surgery, University of Malta, MSD 2090 Msida, Malta; 5Regional Center for Prevention, Diagnosis and Treatment of Medication and Radiation-Related Bone Diseases of the Head and Neck, University of Padova, 35128 Padova, Italy; alberto.bedogni@unipd.it; 6Azienda Ospedaliera SS. Antonio e Biagio e Cesare Arrigo, 15121 Alessandria, Italy; fusco.dott.vittorio@gmail.com

**Keywords:** ONJ, osteonecrosis, treatment, therapy, surgery, staging

## Abstract

Medication-related osteonecrosis of the jaw (MRONJ) is a serious adverse reaction of antiresorptive and antiangiogenic agents, and it is also a potentially painful and debilitating condition. To date, no specific studies have prospectively evaluated the efficacy of its treatment and no robust standard of care has been established. Therefore, a systematic review (2007–2020) with a pooled analysis was performed in order to compare MRONJ surgical techniques (conservative or aggressive) versus combined surgical procedures (surgery plus a non-invasive procedure), where 1137 patients were included in the pooled analysis. A statistically significant difference in the 6-month improvement rate, comparing combined conservative surgery versus only aggressive (91% versus 72%, *p* = 0.05), was observed. No significant difference regarding any group with respect to the 6-month total resolution rate (82% versus 72%) was demonstrated. Of note, conservative surgery combined with various, adjuvant, non-invasive procedures (ozone, LLLT or blood component + Nd:YAG) was found to achieve partial or full healing in all stages, with improved results and the amelioration of many variables. In conclusion, specific adjuvant treatments associated with minimally conservative surgery can be considered effective and safe in the treatment of MRONJ, although well-controlled studies are a requisite in arriving at definitive statements

## 1. Introduction

Medication-related osteonecrosis of the jaw (MRONJ) is a drug-related adverse reaction, characterized by the progressive destruction of bone in the mandible or maxilla. It occurs in subjects currently or previously exposed to treatment with drugs in which an increased risk of osteonecrosis can be observed, in the absence of previous radiation treatment or metastatic disease to the jaw [1]. MRONJ is a severe, emerging orofacial disease, and thousands of cases have been reported since 2003 [2]. It can be associated with two categories of drugs: (1) anti-resorptive agents (e.g., bisphosphonates/BP and denosumab), used for the treatment of diseases involving the skeletal system in patients with bone metastases, hematological (e.g., myeloma) or osteometabolic disease (e.g., primary or secondary osteoporosis, Paget’s disease), and (2) other non-antiresorptive drugs (antiangiogenic agents, mTOR inhibitors, monoclonal antibodies, etc.) for the treatment of different types of cancer. Recent data indicate that the incidence of MRONJ varies between 0 and 12.222 per 100,000 patient-years for cancer patients being treated with intravenous bisphosphonates. The incidence varies from 1.04 to 69 per 100,000 patient-years for osteometabolic patients when taking oral bisphosphonates, whereas this value is 30.2 per 100,000 patient-years [3,4] for patients taking denosumab. This variability depends mainly on several local and systemic risk factors. Regarding the latter, the most recognized are: primary disease in patients (cancer versus osteoporosis), a specific drug regimen, potency of the ONJ-related drug and the cumulative dose of BPs [1,5,6,7]. After a diagnosis of MRONJ, and whilst adhering to clinical and radiological protocols, the stage (i.e., severity and extent of the disease) must be defined in order to provide the patient with effective therapeutic management [7].

Whilst different treatments (therapeutic or palliative) have been described for MRONJ management, it is still a matter of controversy in the oral and maxillofacial communities that a gold standard has not yet been defined. In brief, this standard would involve the three main categories of MRONJ: (a) non-invasive procedures (ranging from pharmacological to laser treatment) [8,9], (b) invasive techniques (i.e., conservative or aggressive surgical approaches) [10] and (c) a combination of (a) and (b) (i.e., surgery plus one of the aforementioned non-invasive procedures) [11]. Non-invasive procedures include: medical treatment, intraoral vacuum-assisted treatment [12], the use of pentoxifylline (associated or not with tocopherol [13,14]), Er:YAG laser ablation, and Nd:YAG/diode laser biostimulation [15,16,17] and teriparatide [18,19,20,21]. Only partial and delayed healing has been reported with non-invasive techniques, to the exclusion of low-level laser treatment (LLLT) and, in certain cases, teriparatide. Furthermore, there is a paucity of high-impact studies in the literature, which would demonstrate effective positive outcomes [22].

Surgical treatments comprise: (i) conservative approaches, such as bone debridement, and sequestrectomy, and (ii) invasive, more aggressive procedures, such as re-sectioning the affected bone and jawbone reconstruction, where indicated. Several studies have yielded very positive results for surgical treatment in MRONJ treatment, especially if performed in the early stages of the disease [23,24,25,26].

Many in the field consider that the term ‘treatment’ is often used inappropriately, in that it is not possible for the disease to heal completely or for the majority of MRONJ patients to arrive at a state of remission. Thus, and as documented in the MRONJ literature, treatment goals are mainly concerned with managing pain, controlling for any infection of the soft and hard tissues and reducing the progression or occurrence of bone necrosis [11]. Over and above every consideration, the authors of this paper hold that maximizing a patient’s quality of life has to be a key feature of every protocol requiring MRONJ treatment.

The aim of this systematic review with a pooled analysis is to examine and compare the main categories of MRONJ treatment: surgical techniques (conservative or aggressive) versus combined procedures (surgery plus non-invasive procedures), by focusing on their therapeutic effectiveness. The recommendations outlined by the Prisma-P 2015 checklist systematic review protocol were followed in order to formulate the methodology for this systematic review.

## 2. Materials and Methods

The present study followed the Preferred Reporting Items for Systematic Reviews and Meta-Analyses (PRISMA) and the Meta-analysis of Observational Studies in Epidemiology (MOOSE) guidelines. The PICO search strategy considered: patients who underwent conservative or aggressive surgery with or without combined procedures (auxiliary treatments). Selected outcomes were a 6-month complete resolution rate, or a 6-month improvement rate in terms of a transition from a higher to a lower stage.

### 2.1. Eligibility Criteria

In order to be included in the systematic review outlined in this paper, studies had to include results from: prospective, non-randomized and randomized clinical trials, retrospective cohort studies and case series (*n* ≥ 10), which investigated the role of surgical (conservative or aggressive) techniques with or without combined procedures (surgery plus a non-invasive one) and with a follow-up ≥ 6 months. Studies were excluded if they constituted a Commentary, Review, Editorial or Protocol. Case series (*n* < 10) or case reports were excluded from the pooled analysis, and the studies were limited to research regarding human beings.

### 2.2. Information Sources and Search Strategy

A systematic, electronic search through different biomedical databases (e.g., PubMed, Ovide/MEDLINE, Web of Knowledge, Embase and the Cochrane Library) was performed by two authors (A.G. and F.C.) in the period from January 2007 to December 2020, and it was restricted to abstracts in English. The MeSH searched terms are included in Table 1.

Furthermore, other data sources (from international meetings and indexed dentistry journals such as Journal of Dentistry, Journal of Oral Maxillofacial Surgery, Journal of Dental Research) were scanned as a source of grey literature.

#### 2.2.1. Study Selection

Screening and eligibility were assessed independently by two reviewers (F.C. and O.D.F.), who were in agreement regarding the results. The Titles of papers and Abstracts were initially screened for relevance and possible eligible results, and thereafter full texts were retrieved. Finally, the reviewers combined their results to create a corpus of selected papers to assess for final eligibility. According to the aim of this review, the resulting papers were allocated to four experimental categories: (1) conservative surgery, (2) aggressive surgery, (3) a conservative plus non-invasive procedure and (4) aggressive surgery plus non-invasive protocols. Table 2 and Table 3 summarize the eligible studies.

#### 2.2.2. Data Collection Process

Data collection was independently performed by two authors (F.C. and A.G.), and their results were reviewed by a third author (O.D.F.) to check for accuracy.

#### 2.2.3. Statistical Analysis

Selected studies were reviewed to detect outcomes of interest. A standard extraction template was used to collect raw data. The extracted data comprised the sample size (*n*), and treatment and clinical outcomes, including: (a) a 6-month complete resolution rate, or (b) a 6-month improvement (in terms of a transition from a higher to a lower stage) rate. Some results were not present in all the trials included in the review. The pooled results for each of the four categories were based on weighted estimates. The I2 statistic test was conducted in order to assess heterogeneity among the included studies. If an I-squared value was lower than 50%, the fixed-effect-based Mantel–Haenszel model was used in order to present variables as weighted measures; if it was higher than 50%, the pooled analysis was performed using the random effect-based model by Der Simonian and Laird. The overall (a) and (b) rates were reported as pooled proportions of the percentage of patients who had been treated with one of the four experimental categories. A two-tailed Student’s t-test with a significance level of 0.05 was used for all comparisons. All analyses were performed using the 15.0 MedCalc for Windows version (MedCalc Software, Ostend, Belgium).

## 3. Results

The initial search strategy identified 371 records, which were obtained by database searching, after the duplicates had been removed. Two reviewers (F.C. and A.G.) independently screened the titles and abstracts to arrive at a total of 121 articles, and 62 duplicates were excluded. Of the 83 articles, 30 did not meet the inclusion criteria for this review, thus 53 articles were eligible (Figure 1). Forty-one out of the fifty-three articles reported only one procedure, and twelve described more than one treatment. Only data from the four relevant categories were extrapolated and included in the pooled analysis. Thus, some studies have been replicated in the distribution among the four categories, taking into account that several authors described different procedures. Authors classified and staged MRONJ cases in all articles in accordance with the guidelines of the American Association of Oral and Maxillofacial Surgeons [27,28]; however, data regarding the extent of the disease was not always available.

Conservative surgery alone: 19 articles were included in this analysis. Even where different surgical approaches had been described, conservative surgery was always preceded by medical treatment (antiseptic mouthwash and systemic antibiotics). Nine out of the nineteen articles were case reports or case series, with a positive outcome in 100% of the treated patients (N = 52, cancer and osteoporotic patients), with a follow-up period of nearly twelve months. Of the remaining 10 articles (N = 375, stages 1, 2, 3 cancer and osteoporotic patients), the overall outcome was successful in the vast majority of cases, recording high success rates of 70% on average with a follow-up period of 9 months (Table 2).

Aggressive surgery alone: 7 articles were included in this analysis. Similarly, the practice of aggressive surgery was always preceded by medical treatment (antiseptic mouthwash and systemic antibiotics). Four of the seven articles were case reports or case series with positive outcomes in 100% of treated patients (N = 25, cancer and osteoporotic patients), with a follow-up period of nearly twelve months. In the 3 remaining articles (N = 245 cancer and osteoporotic patients), the overall outcome was successful in the vast majority of cases, recording high success rates of 70% on average with a follow-up period of 12 months (Table 2).

Conservative surgery plus non-invasive protocol (auxiliary treatment): 36 articles were included in this analysis, 22 of which were case reports or case series (<10 pts) with a positive outcome in 100% of treated patients (N = 51, cancer and osteoporotic patients), with a follow-up period of nearly 11 months. In 15 articles (N = 348, cancer and osteoporotic patients), the overall outcome was successful in almost all cases, recording high success rates of 80% on average with a follow-up period of 8 months. All 36 articles were divided into 13 subcategories on the basis of different auxiliary treatments [29,30,31,32,33,34,35,36,37,38,39,40,41,42,43,44,45,46,47,48,49,50,51,52,53,54,55,56,57,58,59].

Aggressive surgery plus non-invasive procedures (auxiliary treatment): only two papers (case reports) discussed the results of aggressive surgery protocols with auxiliary treatment [49,60].

Table 3 provides a comprehensive overview of these findings of the last categories.

Considering that only studies with *n* > 10 patients were reviewed for the final pooled analysis, a total of 1137 patients were included in the study and assigned to one of the four categories according to their specific intervention. No significant differences between median age and sex were reported as the result of a comparison of the four groups.

The overall 6-month total resolution rate (a) and the 6-month improvement rate (b) were: 74% (CI 95%; 64–83%) and 87% (CI 95%; 78–94%), respectively. The following was reported for (a):80% (CI 95%; 68–90%) for invasive surgery alone (Figure 2a).69% (95% CI; 53–84%) for invasive surgery plus non-invasive procedures (Figure 2b).

The following was reported for (b):81% (CI 95%; 67–92%) for invasive surgery alone (Figure 3a).92% (CI 95%; 88–94%) for invasive surgery plus non-invasive procedures (Figure 3b).

Stratifying for each category of invasive procedures with respect to (a) and to (b), the data are collated in Table 4.

Of interest, a significant statistical difference was observed in the 6-month improvement rate, on comparing combined conservative surgery (mean = 91%) versus only surgical (conservative alone and aggressive alone) techniques (mean 77%, *p* = 0.05). There was no significant difference for any group with respect to the 6-month total resolution rate (82% versus 72%, respectively). No reliable data were available for an analysis of aggressive surgery plus a non-invasive procedure with respect to all the selected indicators.

## 4. Discussion

The most profound effect of MRONJ in patients is the crippling nature of this disease with its negative impact on the quality of life. Thus, the challenge of the medical practitioner in treating these patients is undoubtedly to select the most appropriate medical protocols for maximizing a positive outcome for the patient.

Generally, the management of patients who are unable to achieve *restitutio ad integrum* (total recovery) will imply that the physician accompanies their patients on a journey of chronic illness. However, this may also include the pursuance of innovative research in adhering to appropriate protocols. Nowadays, it is possible for the symptoms of MRONJ to regress to a considerable degree or for the disease to partially heal in the absence of signs of phlogosis and demarcation of the necrotic process. All these features of healing can be considered as the primary goals for MRONJ management [61].

Unfortunately, there currently lacks consensus regarding the most appropriate treatment strategy for MRONJ. This may, in part, be due to the heterogeneity of MRONJ staging and available treatments; indeed, the majority of proposed protocols are surgical (conservative or aggressive). A conservative surgical approach can be deployed with the debridement of the superficial necrotic bone or by sequestrectomy, in addition to the use of systemic antibiotics and local antiseptic products [62,63,64,65,66,67,68,69]. An aggressive surgical approach will be adopted when conservative debridement has failed; in advanced cases of MRONJ, it is characterized by the partial or total resecting of the necrotic/non-necrotic tissue [70,71,72].

Two parameters, which are considered fundamental in deciding how to treat MRONJ, are staging and healing. A well-established staging system should be used to quantify the severity and extent of MRONJ and to guide decision-making [73]. Nowadays, AAOMS staging, as it is often termed, does not always satisfy the requisites for MRONJ treatment since it does not take into account the effective anatomical/radiological extent of the disease [28]. In this regard, Campisi et al. have recently highlighted this limitation, concluding that it may not always be effective and appropriate to propose MRONJ treatments based on the staging system [74]. The trials relating to their research, included in this systematic review, do not contain data stratified by disease stage, thereby precluding a pooled evaluation on this key topic. With the aim of facilitating reading, the authors of this paper have inserted the staging system used in every case series in order to complete the descriptive data (Table 1 and Table 2). Of note, the choice of a given surgical protocol in some cases was found to be independent of the staging evaluation.

Healing is another pillar of MRONJ management. Based on an assessment of symptomatology, and clinical mucosal and radiological signs, specific terms have recently been proposed to improve the descriptions of treatment outcomes. Examples include: “resolved”, “improving”, “stable” and “progressive” [73]. Unfortunately, most academic papers have described the process of healing by merely referring to the integrity of mucosa as being seamless and symptomless, however without evaluating the underlying bone, from which the disease develops and recurs [75]. Furthermore, follow-up periods are often very short (approximately 6 months), and they exclude a recurrence at 6–12 months after the surgical procedure [6,76]. The analysis proposed herein has described healing as complete or partial as the authors have limited their descriptions to using undefined data.

Referring to the main results of this systematic analysis, high success rates (70% on average) with a follow-up period of 9 months were recorded in almost all treated patients (N = 375) who had been treated solely according to a conservative surgical approach. Approximately 70% of the 270 MRONJ patients, who had been treated solely with aggressive surgery and a follow-up of more than 12 months, also had a successful MRONJ outcome. Of this latter group, aggressive surgery alone was deployed in stage 1 of the disease [23,65,77]. This surgical choice is not supported by statements or guidelines, but it is left to the discretion of the surgeon; it does not, therefore, permit a reliable analysis of the results.

In a dissimilar manner, a non-invasive procedure (auxiliary therapy) was combined with conservative surgery, whilst there are fewer cases of aggressive surgery reported in the literature. When conservative surgery was combined with auxiliary therapy in the management of 401 patients, the optimum patient outcome was achieved by only considering the data of partial healing, and no differences have been reported in the literature when complete resolution (healing) was evaluated (Figure 2 and Figure 3).

Auxiliary therapy consists of several agents with antiseptic, angiogenic and biomodulatory properties. Of these, platelet-rich fibrin (PRF) is a second-generation platelet concentrate. PRF membranes appear to be an easy and cheap alternative treatment with which to close exposed bone in MRONJ after surgical treatment [78,79,80,81]. They promote gingival healing by acting as a barrier membrane between the alveolar bone and the oral cavity, thereby accelerating physiologic wound healing and new bone formation. This minimizes recurrent infections and it prevents osteonecrotic lesions [34,35,36,38,50,82]. Moreover, leukocyte and platelet-rich fibrin (L-PRF and PRF, respectively) are determinant in immune regulation, which may be of importance in reducing tissue infection in the immediate postoperative period [83,84,85]. The association between surgery and blood components (such as platelet-rich—PRP or PRF or L-PRF) in the analysis presented in this paper has proved itself to be an effective treatment for the rapid closure of bone exposure and the formation of new gingival tissue in the absence of signs of phlogosis. With a success rate of 95%, several authors have demonstrated that this association is effective on average over a 14-month follow-up period.

Another safe and effective treatment has also been described in the literature: PRF and PRP have been associated with laser phototherapy by virtue of the elevated microbicidal activity of the laser [16,43], making use of autologous bone marrow stem cells. This research has reported effective results by virtue of the osteogenic and chondrogenic potential of these agents (PRF and PRP), in addition to their capacity for vasculogenesis and angiogenesis [43]. De Santis et al [86] described two case reports treated with the debridement of the exposed necrotic bone followed by bone marrow stem cells injection: a positive follow up with no sing or symptoms in the necrotic area has been reported for the next 13 months. 

Referring to the systematic review described herein, the associations between conservative surgery plus blood components, and laser or photodynamic therapy, appear to contribute much to: newly formed bone, the full coverage of bone tissue with healthy mucosa and the absence of symptoms and other signs of necrotic progression. This is due to the analgesic, anti-inflammatory and biomodulatory effects of blood components, and this protocol has been shown to be effective on average over a 6-month follow-up period with a success rate of 86%.

The association of autologous bone marrow stem cells with conservative surgery and blood components has been reported only in one case study, with a success rate of 100% on average over a 6-month follow-up period. The CT scan revealed the diminution of osteolytic lesions with complete bone regeneration of the medial cortex of the lower jaw and a total resolution of symptoms.

An early resolution of MRONJ has been reported when combining conservative surgery and the use of bone morphogenetic protein (BMP-2) or SVC (stromal vascular fraction: a heterogeneous cell population containing mesenchymal stromal cells isolated by adipose tissue) and L-PRF: the addition of BMP to L-PRF stimulates soft tissue healing and bone remodeling, thereby promoting total mucosal coverage in the absence of signs of phlogosis and exposed bone, leading to a marked diminution of symptoms. Thus, patients requiring the continued administration of antiresorptive treatment may benefit from such a combined regimen [31,87].

Moving on to another study, Jung et al. [42] have proposed a concomitant administering of BMP-2 with teriparatide (TPTD) in order to maximize the regeneration of bone after surgery. In a synergistic manner, TPTD stimulates an anabolic effect by accelerating the osteoblastic differentiation of the BMP [88]. This result may prompt a paradigm shift in the treatment of MRONJ from resecting to regeneration. The association between BMP and TPTD had a success rate of 100% over a 3-month follow-up period. The addition of TPTD to BMP enhances remodeling and the formation of bone, thereby facilitating healing and the removal of necrotic bone. Many patients experience a complete resolution of their symptoms with no signs of phlogosis.

Low-level laser therapy (LLLT) has been used in treating MRONJ patients, together with an AF-guided surgical or a conventional surgical approach. It is considered by many as a safe and effective adjunct to the medical-surgical treatment of MRONJ lesions because it stimulates the regeneration and angiogenesis of soft tissues, thereby increasing the duration of the healing process. However, there still exists controversy regarding the physical and biological variables of LLLT, including: the type of laser, the frequency of the light pulse, output power, duration of application, fluence, and the distance of the source from the irradiated tissue [45,46,47,48,49]. The association of LLLT with surgery has demonstrated a success rate of 87% on average over a 12-month follow-up period, with total mucosal healing in the absence of signs of infection or pain.

The use of surgery has also been associated with teriparatide (TPTD) treatment (prior to or after conventional surgical treatment) for MRONJ in osteoporotic patients. TPTD stimulates trabecular and cortical thickness, and trabecular connectivity and bone size bone formation by increasing osteoblast number and activity. Although successful results using TPTD treatment have been reported in the literature, its safety and efficacy are currently awaiting comprehensive evaluation. The treatment time during which it can be safely administered is strictly limited to less than 2 years in one lifespan [57,58,59]. A success rate of 83% on average over an 11-month follow-up period has been reported for the surgical treatment plus TPTD treatment (or vice versa) of MRONJ: any surgical wound completely healed with X-rays indicating stable alveolar bone. No inflammatory signs and symptoms have been reported to date.

Other protocols (for example, the use of ozone and hyperbaric oxygen (HBO)) have also been deployed and evaluated as a MRONJ surgical adjuvant treatment. Ozone has been used with different formulations (i.e., an oral irrigation of aqueous ozone, gas insufflation) and duration (prior to or after surgical curettage or sequestrectomy) by virtue of its positive features. These include: antimicrobial power, an enhancement of tissue oxygenation, an activation of the immune response, a stimulation of angiogenesis and fibroblast formation and analgesic agents [51,55,56,89]. In this review, the association between ozone and conservative surgery (or vice versa) demonstrated a success rate of 90–100% on average over a 22-month follow-up period. Complete mucosa healing was seen in the absence of symptoms such as pain and local inflammation.

As a pre-surgical treatment, HBO has successfully treated MRONJ lesions, thereby: improving the quality of life of afflicted patients [52,53,54], increasing wound healing, and reducing edema, inflammation and pain. HBO followed by surgical treatment had a success rate of 84% on average over an 18-month follow-up period, with: the complete healing of MRONJ lesion, total mucosal coverage, a cessation in the signs of infection and notable symptomatic relief.

## 5. Conclusions

The authors of this paper performed an evidence-based analysis which demonstrated the compelling and effective performance of non-invasive procedures, combined with conservative and aggressive surgery, in treating MRONJ patients. The data confirmed that partial and complete 6-month resolution rates ranged from 70% to 100%. Of note, adjuvant therapy usually requires daily or weekly applications. Such a regular clinical practice permits the surgeon to constantly monitor the MRONJ lesion and to promptly modify treatment, where indicated. It is also hoped that many patients will be more inclined to maintain effective oral hygiene on account of their continual checkups. Moreover, an alleviation in symptoms has been achieved using appropriate treatments within a relatively short period of time and in the absence of negative events.

Many MRONJ patients can achieve total remission by means of aggressive surgical treatment, which is similar in mean duration to conservative surgery alone (9–12 months). This is notwithstanding other considerations, such as the patient suffering from a debilitating disease, the exacerbation of the quality of life with marked morbidity and, last but not least, challenging conditions for patients after aggressive surgery. These factors must be taken into account if patients also suffer from a significant systemic disease (e.g., metastatic patients) [41]. It might be opportune to highlight in cancer patients the appropriate choice of an MRONJ management protocol by conservative surgery with the addition of ozone [51], LLLT [45] or blood component + Nd:YAG [47] laser treatment. The analysis in this paper has demonstrated improved results in treating MRONJ with nearly total healing. Regrettably, there is a lack of reported data relating to the use of aggressive surgery plus auxiliary protocols, which would have been included in the pooled analysis.

Finally, some studies discussed in this paper confirmed an extended follow-up period for patients. This represents a key point in evaluating the healing of MRONJ, as has been previously highlighted by two Italian scientific bodies of oral pathology and medicine, and maxillofacial surgery: SIPMO and SICMF, respectively [74]. Their research has defined clinical and radiological MRONJ healing with a documented absence of symptoms and the clinical signs of MRONJ in a period of no less than 12 months [1,3,90].

Despite the systematic nature of the analysis in this paper, there are limitations relating to: the non-randomized retrospective/prospective nature of the studies herein, the analysis of historical data, the heterogeneity of patients included in the study and a suitable definition of the endpoints being examined (the complete and partial resolutions of MRONJ symptoms).

To the best of our knowledge, the following regarding MRONJ treatment can be highlighted: 

(1) A unanimous factual definition, including evaluation criteria (diagnosis and staging), is fundamental in assessing the efficacy of well-specified MRONJ treatment in order to facilitate a systematic analysis of the results of the research.

(2) The main positive outcomes of MRONJ treatment should be: the absence of symptoms, clinically active phlogosis and the obstacle of the relevant area of bone, as recognized on CT scans for a period of at least 12 months. 

(3) Many in the field would say that the treatment of MRONJ is unquestionably related to its staging and the systemic status of the patient: cancer patients have often the worst quality of life, and aggressive surgery can exacerbate their condition. 

(4) Conservative surgery combined with adjuvant procedures (i.e., ozone, LLLT or blood component + Nd:YAG laser treatment) can contribute to partial or total healing in all stages of MRONJ, with improved results and variables (from symptoms to clinical and radiological signs).

(5) Adjuvant therapy associated with surgery (conservative or aggressive) may be the future for MRONJ treatment. This combination could lead to the most positive results, but it is also of the utmost importance for conducting further effectively controlled studies in order to arrive at conclusive statements for the effective treatment of MRONJ.

## Figures and Tables

**Figure 1 ijerph-18-08432-f001:**
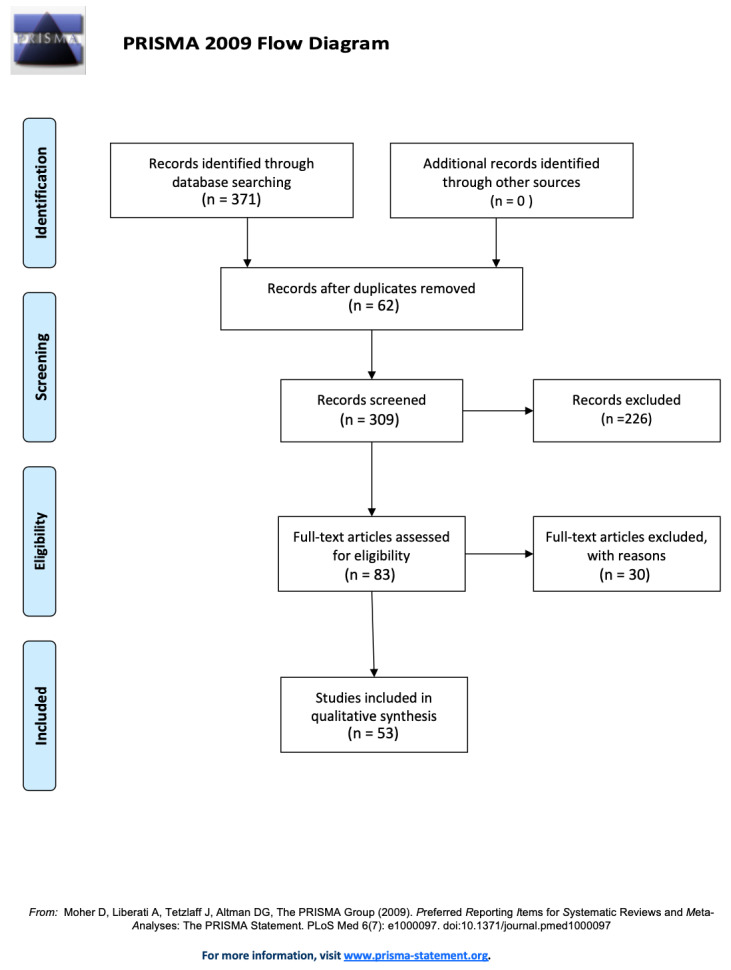
A PRISMA flow chart of the pooled studies.

**Figure 2 ijerph-18-08432-f002:**
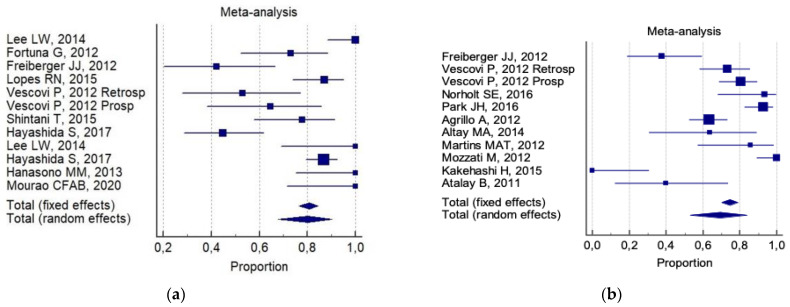
Forest plot results of pooled results about complete resolution in (**a**) invasive (conservative/aggressive) treatments, and (**b**) invasive (conservative/aggressive) treatments + non-invasive treatments.

**Figure 3 ijerph-18-08432-f003:**
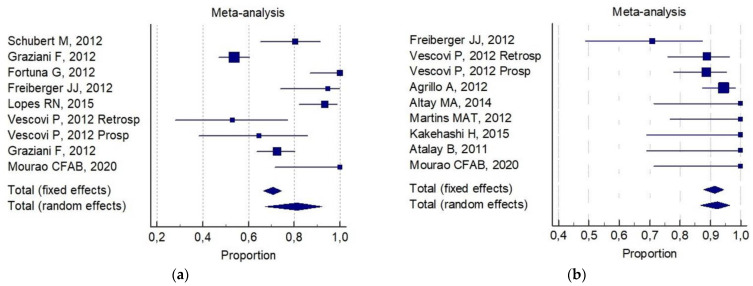
Forest plot results of pooled results about complete resolution in (**a**) invasive (conservative/aggressive) treatments, and (**b**) invasive (conservative/aggressive) plus non-invasive treatments.

**Table 1 ijerph-18-08432-t001:** MeSH terms.

“Osteonecrosis”[Mesh] AND “Jaw Diseases”[Mesh]
▪ AND (“Conservative Treatment”[Mesh]
○ OR “Drug Therapy”[Mesh]
○ OR “Therapeutics”[Mesh]
○ OR “therapy” [Subheading]
○ OR “Surgical Procedures, Operative”[Mesh]
○ OR “drug therapy” [Subheading])
▪ AND “Ozone”[Mesh]
▪ AND “Teriparatide”[Mesh]
▪ AND (“Laser Therapy”[Mesh]
○ OR “Low-Level Light Therapy”[Mesh])
▪ AND “Pentoxifylline”[Mesh]
▪ AND “Hyperbaric Oxygenation”[Mesh]
▪ AND “Tocopherols”[Mesh]
▪ AND “Platelet-Rich Plasma”[Mesh]
▪ AND “Bone Morphogenetic Proteins”[Mesh]
▪ AND “Parathyroid Hormone”[Mesh])

**Table 2 ijerph-18-08432-t002:** Summary of the characteristics and the results of the studies concerning MRONJ surgical therapies.

Treatment	Study	Study Type	Pts	Intervention	Outcome	Follow-Up
**Conservative Surgery**	De Souza Povoa et al., 2016	Case report	N = 1 Onc Stage 1	Removal of the exposed necrotic bone and primary wound closure	Complete healing and new bone formation in the surgical site	26 months
	Ribeiro et al., 2015	Case report	N = 1 Ost Stage unspecified	Surgical removal of whole necrotic bone, extraction of all compromised teeth	Complete healing	12 months
	De Souza Faloni et al., 2011	Case report	N = 1 Ost Stage 2	Conservative debridement of the necrotic bone and of part of the surrounding healthy bone, as a margin of safety	Complete healing	8 months
	Pechalova et al., 2011	Case series	N = 3 Onc Stage unspecified	Conservative surgical debridement	Complete healing	Average of 4 months
	Martins et al., 2012	Retrospective clinical study	N = 5 Onc Stage 1,2	Sequestrectomy and/or ostectomy and/or osteoplasty until bone marrow bleeding	60% patients completely healed	6 months
	Jung et al., 2017	Case series	N = 7 Ost Stage 2,3	Patient underwent conventional surgery, and the bone defects were filled with absorbable collagen plugs.	Complete healing and new bone formation in the surgical site	3 months
	Atalay et al., 2011	Retrospective clinical study	N = 10 Onc Stage	The affected bony tissues were curetted from the surface of the bone using bone curettes and round tungsten carbide burs. The necrotic bone was completely removed until the vital bone tissues and vessel spots appeared	40% patients completely healed	6 months
	Vescovi et al., 2012	Retrospective clinical study	N = 17 Onc + Ost Stage 1,2,3	Conservative surgical treatments consisted of sequestrectomy of necrotic bone, superficial debridement/curettage, or corticotomy/surgical removal of alveolar and/or cortical bone	53% patients completely healed	9 months
	Vescovi et al., 2011	Prospective clinical study	N = 17 Onc + Ost Stage 1,2,3	Conservative surgical treatments included sequestrectomies, superficial debridement/curettage and corticotomies/surgical removal of surrounding alveolar and/or cortical bone	65% patients completely healed	12 months
	Freiberger et al., 2012^5^	Randomized control trial	N = 19 Onc + Ost Stage 1,2,3	Surgical debridement of the necrotic bone	33% patients completely healed	24 months
	Fortuna et al., 2012	Single-center prospective open-label clinical trial	N = 26 Onc Stage 2,3	Systemic and topical antibiotic therapy following by sequestrectomy	73% patients completely healed	Average of 10 months
	Lee et al., 2014	Case series	N = 30 Ost + Onc Stage 1,2,3	Minor surgical debridement was performed after irrigation, in which the necrotic bone fragments were removed	Complete healing	Average of 16 months
	Schubert et al., 2012	Prospective study	N = 54 Onc + Ost Stage 1,2,3	Complete electrical or manual removal of the osteonecrosis until points of bleeding from the bone can be macroscopically detected.	88.8% patients completely healed	6 months (72%)
	Graziani et al., 2012	Retrospective cohort multicenter study	N = 227 Ost + Onc Stage 1,2,3	Local debridement was comprised of all surgical interventions, such as sequestrectomy, soft tissue debridement and curettage, that did not require bone surgery beyond the regular margins	49% patients completely healed	6 months
**Conservative Surgery with Buccal Fat Pad Closure**	Duarte et al., 2015	Case report	N = 1 Onc Stage 2	The extensive necrotic bone area was surgically removed, resulting in oral sinus communication. A buccal fat pad was used to cover the defect	Complete healing	3 months
	Gallego et al., 2012	Case series	N = 3 Onc + Ost Stage 1,2,3	Sequestrectomy and bone debridement. The overlying mucosa was sutured over the defect with reconstruction with buccal fat pad.	Complete healing	Average of 12 months
	Berrone et al., 2015	Case series	N = 5 Onc Stage 3	Removal of the necrotic bone and primary closure of the oroantral communication using a buccal fat pad flap.	Complete healing	Average of 12 months
	Lopes et al., 2015	Retrospective observational cohort study	N = 46 Onc + Ost Stage 2,3	Removal of all necrotic bone until bleeding was obtaining at the bony margins, conscious smoothing of all sharp bone edges and primary closure of the wound.	87% patients completely healed	10 months
	Hayashida et al., 2017	Multicenter retrospective study	N = 38 Onc + Ost Stage 1,2,3	One group received conservative surgery, removal of only the necrotic bone and extensive surgery, defined as removal of the necrotic and surrounding bone (marginal mandibulectomy or partial maxillectomy).	76.7% patients completely healed	Average of 15 months
**Aggressive Surgery**	Hewson et al., 2012	Case report	N = 1 Onc Stage 3	Radical surgical excision of all diseased bone and nasio-labial flap reconstruction.	Complete healing	6 months
	Ghazali et al., 2013	Case report	N = 1 Ost Stage 3	Hemimandibulectomy and an osteocutaneous fibula flap reconstruction	Complete healing	24 months
	Shintani et al., 2015	Cohort study	N = 4 Ost + Onc Stage 1,2,3	Segmental resection and immediate reconstruction with a reconstruction plate were performed.	3/4 patients completely healed	12 months
	Lee et al., 2014	Case report	N = 10 Ost + Onc Stage 1,2,3	Large necrotic bone segment was removed by an ultrasonic bone saw. A bone file or rongeur was used for rounding the sharp bone edge. Then, the bone defect was closed by sutures or COE pack.	Complete healing	Average of 8 months
	Hanasono et al., 2013	Case series	N = 13 Onc Stage2, 3	Segmental mandibulectomy and microvascular free flap reconstruction.	Complete healing	Average of 15 months
	Graziani et al., 2012	Retrospective cohort multicenter study	N = 120 Ost + Onc Stage 1,2,3	Re-sective procedures were defined as corticotomy, surgical removal of the lesion and extended bone removal without prejudice for the continuity of the mandible/maxilla.	68% patients completely healed	6 months
	Hayashida et al., 2017	Multicenter retrospective study	N = 121 Onc + Ost Stage 1,2,3	Extensive surgery, defined as removal of the necrotic and surrounding bone (marginal mandibulectomy or partial maxillectomy).	86.8% patients completely healed	Average of 15 months

**Table 3 ijerph-18-08432-t003:** Summary of the characteristics and the results of the studies on MRONJ surgery plus non-invasive procedures.

	Study	Study Type	Population	Intervention	Outcome	Follow-Up
**Conservative surgery plus (+) non-invasive procedures**						
**1. Surgery + Blood Component**	Gönen et al., 2017	Case report	N = 1 Onc Stage 3	Sequestrectomy + PRF	Complete resolution	18 months
	Soydan et al., 2014	Case report	N = 1 Onc Stage unspecified	Curettage + PRF	Complete resolution	6 months
	Maluf et al., 2016	Case series	N = 2 Onc Stage 2	Resection of the necrotic tissues, curettage and osteotomy + L-PRF	Partial healing	6 months
	Dincă et al., 2014	Retrospective clinical study	N = 10 Onc Stage 2	Sequestrectomy or curettage + PRF	Complete resolution	1 month
	Nørholt et al., 2016	Prospective study	N = 15 Onc + Ost Stage 2,3	Curettage + L-PRF	93.3% patients completely healed	20 months
	Anitua et al., 2013	Case report	N = 1 Onc Stage unspecified	Curettage + PRGF	Complete resolution	12 months
	Bocanegra-Pérez et al., 2012	Prospective descriptive study	N = 8 Onc + Ost Stage 2	Curettage + PRP	Complete resolution	14 months
	Mozzati et al., 2012	Retrospective clinical study	N = 32 Onc Stage 2	Conservative surgery + PRFG	Complete resolution	From 48 to 50 months
	Tsai et al., 2016	Case report	N = 1 Ost Stage 3	Surgical debridement, sequestrectomy + PRF	Complete resolution	10 months
	Pelaz et al., 2014	Cohort study	N = 5 Ost Stage 3	Sequestrectomy and curettage + PRF	Complete resolution	An average of 20 months
	Park et al., 2017	Prospective study	N = 25 Onc + Ost Stage 1,2,3	Conservative surgery + L-PRF	36% patients completely healed	4 months
	Fernando de Almeida Barros Mourao C et al., 2020	Case series	N = 11 Ost Stage 2	Surgical removal of necrotic bone + PRF membranes	Complete healing	24 months
	Giudice A et al., 2020	Case report	N = 1 Ost Stage 3	Surgical removal of necrotic bone + PRF membranes	Complete healing	60 months
	Bouland C et al., 2020	Case report	N = 2 Ost + Onc Stage 2 and 3	Surgical removal of necrotic bone + SVF and L-PRF membranes	Complete healing	18 months
**2** **. Surgery + Blood Component + Photodynamic Therapy**	De Castro et al., 2016	Case series	N = 2 Ost Stage 2,3	Surgical debridement + PDT + PRF	Complete resolution	An average of 12 months
**3. Surgery + Blood Component +** **Bone Morphogenetic Protein**	Park et al., 2017	Prospective study	N = 30 Onc + Ost Stage 1,2,3	Conservative surgery + combined L-PRF and recombinant human BMP-2 (rhBMP-2)	60% patients completely healed	4 months
**4. Surgery + Teriparatide**	Lee et al., 2010	Case report	N = 1 Ost Stage 2	Sequestrectomy + teriparatide	Complete resolution	6 months
**5. Surgery + Teriparatide + Bone Morphogenetic Protein**	Jung et al., 2017	Cohort study	N = 6 Ost Stage 2,3	Conservative surgery and absorbable collagen plugs soaked by rhBMP-2 into the bone defect plus daily subcutaneous injection of 20 mg teriparatide for 1–4 months.	Complete resolution	3 months
**6. Surgery + Bone Morphogenetic Protein**	Jung et al., 2017	Cohort study	N = 4 Ost Stage 2,3	Conservative surgery and absorbable collagen plugs soaked by rhBMP-2 into the bone defect.	Complete resolution	3 months
**7. Surgery + Blood Component + Autolugus Bone Marrow Stem Cells**	Gonzálvez-García et al., 2013	Case report	N = 1 Onc Stage 2	Removal of the necrotic bone+ bone marrow stem cells + beta tricalcium phosphate + demineralized bone matrix + PRP	Complete resolution	6 months
	De Santis et al., 2020	Case report	N = 2 Onc Stage 2	Debridement of the exposed necrotic bone followed by bone marrow stem cells injection	Complete healing and new bone formation in the surgical site.	13 months
**8. Surgery + LLLT**	Da Guarda et al., 2012	Case report	N = 1 Onc Stage unspecified	GaAlAs diode laser every 48 h for 10 days + antibiotic therapy + curettage	Complete resolution	6 months
**9. Surgery + Blood Component + Laser Phototherapy**	Altay et al., 2014	Retrospective clinical study	N = 11 Onc Stage2,3	Pre- and post-operative antibiotic administrations + GaA-lAs diode laser	Complete resolution	12 months
	Atalay et al., 2011	Retrospective clinical study	N = 10 Onc Stage 1,2	Conservative surgery + low-level laser therapy application (Er:YAG and Nd:YAG)	70% patients completely healed	12 months
	Vescovi et al., 2012	Retrospective clinical study	N = 45 Onc + Ost Stage 1,2,3	Conservative surgery + laser Nd:YAG	89% patients completely healed	6 months
	Vescovi et al., 2011	Prospective clinical study	N = 62 Onc + Ost Stage 1,2,3	Conservative surgery + laser LLLT	73% patients completely healed	17 months.
	Martins et al., 2012	Retrospective clinical study	N = 14 Onc Stage 1,2,3	Conservative surgery + continuous indium-gallium-aluminum-phosphide diode laser. The LPT treatment started on the first visit and continued daily until mucosal healing was observed.	86% patients completely healed	12 months
**10. Surgery + Ozone**	Agrillo et al., 2012	Retrospective study	N = 94 Onc + Ost Stage unspecified	Curettage or sequestrectomy + Ozone therapy (3 min sessions 2/week) + pharmacological therapy	90% patients completely healed	An average of 6 months
**11. HBO + Surgery ***	Fatema et al., 2013	Case report	N = 1 Onc Stage 2	Antibiotics therapy, irrigation, pre-operative HBO therapy for 20 sessions, conservative minor surgical debridement and again post-operative HBO therapy for ten sessions.	Complete resolution	Unspecified
	Al-Zoman et al., 2013	Case series	N = 3 Onc Stage2,3	HBO therapy, oral/parenteral antibiotic, analgesics, conservative surgery (debridement of bone sequestra) and daily rinsing with chlorhexidine mouthwash.	Complete resolution	12 months
	Freiberger et al., 2012	Randomized control trial	N = 24 Onc + Ost Stage 1,2,3	40 HBO treatments at 2.0 atm for 2 h twice per day and conservative surgical debridement of the necrotic bone.	52% patients completely healed	24 months
**12. Ozone + Surgery** *****	Ripamonti et al., 2012	Case report	N = 1 Onc Stage unspecified	Antibiotic + antimycotic therapy for 10 days. Local ozone gas (total of 15 applications). Conservative surgery (sequestrectomy).	Complete resolution	36 months
	Brozoski et al., 2020	Case series	N = 2 Onc + Ost Stage 2	Weekly irrigation with aqueous ozone solution on bone-exposed region + daily mouthwashes of ozone solution. After 3 and 6 months: conservative surgery (debridement and sequestrectomy)	Complete resolution	An average of 24 months
**13. Teriparatide + Surgery ***	Doh et al., 2015	Case report	N = 1 Ost Stage 2	After 4 months of daily teriparatide therapy conservative surgery (sequestrectomy). The TPTD therapy was terminated 6 months after the initial treatment.	Complete resolution	20 months
	Kwon et al., 2012	Case series	N = 6 Ost Stage 2,3	Daily Teriparatide (20 μg/day) 1–3 months + conservative sequestrectomy/marginal/aggressive segmental resection	Complete resolution	3 months
	Kakehashi et al., 2015	Case series	N = 10 Ost Stage 2,3	Daily teriparatide (20 μg/day) ranged from 4 to 24 months. In some cases, surgery was performed to obtain the healing.	Partial resolution	From 4 to 24 months (duration of teriparatide therapy until mucosal healing)
**Aggressive surgery plus non-invasive procedures**						
**1. Surgery + Bone Graft + Bone Morphogenetic Protein**	Rahim I 2015	Case report	N = 1 Ost Stage 3	Partial mandibulectomy + bone graft from the iliac crest + rhBMP-7	Complete resolution	60 months
**2. AF-Guided Surgery + LLLT**	Vescovi P 2015	Case report	N = 1 Onc Stage 3	Osteotomy with Er:YAG laser + AF visualization to guide the osteoplasty. Intraoral irrigations with povidone iodine solution + application of Nd:YAG laser + weekly applications of LLLT for 3 weeks after intervention	Complete resolution	7 months

* Procedures administered prior to surgery.

**Table 4 ijerph-18-08432-t004:** Stratification for each category of invasive procedures with respect to (a) and to (b).

	6-Month Total Resolution Rate (a)	6-Month Improvement Rate (b)
Conservative surgery alone	67% (IC 95%; 50–83%)	82% (IC 95%, 65–95%)
Aggressive surgery alone	93% (IC 95%; 82–99%)	72% (IC 95%; 64–80%)
Conservative surgery plus non-invasive procedures	75% (IC 95%; 60–87%)	91% (IC95%; 87–96%)
Aggressive plus non-invasive procedures	not assessable	not assessable

## Data Availability

No new data were created or analyzed in this study. Data sharing is not applicable to this article.

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
