# Peer review of "The Treatment of Medication-Related Osteonecrosis of the Jaw (MRONJ): A Systematic Review with a Pooled Analysis of Only Surgery versus Combined Protocols"

_ijerph, 2021, doi:10.3390/ijerph18168432_

Round 1

Reviewer 1 Report

The aim of authors was to compare the effects of surgery in the abscence or presence of adjuvant therapeutic approaches in patients with MRONJ.

The manuscript summarize the current knowledge about MRONJ and its management. However, this severe complication is defined erroneously in the first paragraph, since MRONJ can be associated not only with antiresorptive and antiangiogenic drugs, but also with non-antiresorptive drugs (e.g. mTOR inhibitors or other monoclonal antibodies). Material and methods are adequate, and based on the PRISMA and MOOSE guideline. The Mesh terms involve only bisphosphonate-associated osteonecrosis of the jaw, although, the aim was to examine and compare the main categories of MRONJ treatment. Publications were collected from the period of January 2007 to December 2019, however, to date some novel, relevant articles have been also published in this topic. Some relevant articles were also out of Discussion (e.g. Govaerts D et al. Adjuvant therapies for MRONJ: A systematic review. 2020 or de Souza Tolentino E et al. Adjuvant therapies in the management of medication-related osteonecrosis of the jaws: Systematic review. 2019.), where the authors could compare their results with others.

In my opinion, the investigated period, the Mesh terms and the Discussion should be expanded, as suggested above. 

Author Response

The aim of authors was to compare the effects of surgery in the abscence or presence of adjuvant therapeutic approaches in patients with MRONJ.

The manuscript summarize the current knowledge about MRONJ and its management. However, this severe complication is defined erroneously in the first paragraph, since MRONJ can be associated not only with antiresorptive and antiangiogenic drugs, but also with non-antiresorptive drugs (e.g. mTOR inhibitors or other monoclonal antibodies).

At the best of our knowledge, other drugs such as mTOR inhibitors or other monoclonal antibodies have also anti-angiogenic activity. In text, we modify  in “antiresorptive drugs (bisphosphonates, densoumab) and other non-antiresorptive drugs (antiangiogenic agents; mTOR inhibitors; monoclonal antibodies; etc)”- .lines 48-50

Material and methods are adequate, and based on the PRISMA and MOOSE guideline. The Mesh terms involve only bisphosphonate-associated osteonecrosis of the jaw, although, the aim was to examine and compare the main categories of MRONJ treatment.

Yes, sorry, it is a mistake of our study at the beginning and now table 1 has been updated and corrected.

Publications were collected from the period of January 2007 to December 2019, however, to date some novel, relevant articles have been also published in this topic. Some relevant articles were also out of Discussion (e.g. Govaerts D et al. Adjuvant therapies for MRONJ: A systematic review. 2020 or de Souza Tolentino E et al. Adjuvant therapies in the management of medication-related osteonecrosis of the jaws: Systematic review. 2019.), where the authors could compare their results with others.

Done

In my opinion, the investigated period, the Mesh terms and the Discussion should be expanded, as suggested above. 

DONE

Please see modified manuscript attached

Reviewer 2 Report

Dear authors congratulations for your article, it is very interesting and full of clinical observations. I added only few suggestions to make it published according to my scientific opinion.

Abstract:
To make more clear the results in the abstract I suggest you to add the information of table 4 in the text of the abstract.
INTRODUCTION:

Please go in deeper about the role of ozone and other medication as a possible help in the healing phases. Probably they could be a good co-adjuvant therapy with the conservative surgery, but you should explicate the rationale of this therapies in osteonecrosis. I suggest to read this case report, but you do not have to cite it because there are more important and valuable review on the ozone in dentistry:
Mandibular Osteonecrosis Associated with Antacid Therapy (Esomeprazole). Eur J Case Rep Intern Med. 2019 Oct 14;6(10):001279. doi: 10.12890/2019_001279.

Line 93-96: You stated that you followed the PRISMA guidelines. If you have developer the systematic review protocol and registered at PROSPERO (National Institute for Health Research, University of York, York, UK) please report the code number.

Line 327-331: In this sentence to validate what you say, please cite this following classic animal study that is focused on the bone healing influenced by the soft tissue healing.
Marconcini S, Denaro M, Cosola S, Gabriele M, Toti P, Mijiritsky E, Proietti A, Basolo F, Giammarinaro E, Covani U. Myofibroblast Gene Expression Profile after Tooth Extraction in the Rabbit. Materials (Basel). 2019 Nov 9;12(22):3697. doi: 10.3390/ma12223697. 

Author Response

Abstract:
To make more clear the results in the abstract I suggest you to add the information of table 4 in the text of the abstract. DONE- LINES 26-30

INTRODUCTION:

Please go in deeper about the role of ozone and other medication as a possible help in the healing phases. Probably they could be a good co-adjuvant therapy with the conservative surgery, but you should explicate the rationale of this therapies in osteonecrosis. I suggest to read this case report, but you do not have to cite it because there are more important and valuable review on the ozone in dentistry:

Other auxiliary therapies, including several agents with antiseptic, angiogenic and biomodulatory properties have discussed from line 297.

The role of ozone has been reported in discussion-  lines 366-381

Mandibular Osteonecrosis Associated with Antacid Therapy (Esomeprazole). Eur J Case Rep Intern Med. 2019 Oct 14;6(10):001279. doi: 10.12890/2019_001279.-

DONE

Line 93-96: You stated that you followed the PRISMA guidelines. If you have developer the systematic review protocol and registered at PROSPERO (National Institute for Health Research, University of York, York, UK) please report the code number.

Submitted; it is being assessed by the editorial team (pdf in attachment)

Line 327-331: In this sentence to validate what you say, please cite this following classic animal study that is focused on the bone healing influenced by the soft tissue healing.
Marconcini S, Denaro M, Cosola S, Gabriele M, Toti P, Mijiritsky E, Proietti A, Basolo F, Giammarinaro E, Covani U. Myofibroblast Gene Expression Profile after Tooth Extraction in the Rabbit. Materials (Basel). 2019 Nov 9;12(22):3697. doi: 10.3390/ma12223697. –

DONE

Round 2

Reviewer 1 Report

The authors corrected the manuscript as required. In this form is appropriate for publictaion.